# Bioelectric Impedance Vector Analysis (BIVA) in Breast Cancer Patients: A Tool for Research and Clinical Practice

**DOI:** 10.3390/medicina55100663

**Published:** 2019-09-30

**Authors:** Ana Teresa Limon-Miro, Mauro E. Valencia, Veronica Lopez-Teros, Alan Eduardo Guzman-Leon, Herminia Mendivil-Alvarado, Humberto Astiazaran-Garcia

**Affiliations:** 1Department of Nutrition, Centro de Investigacion en Alimentacion y Desarrollo A.C. (CIAD), Carretera Gustavo Enrique Astiazaran Rosas 46, Hermosillo 83304, Mexico; analimonmiro@gmail.com (A.T.L.-M.); mauro@ciad.mx (M.E.V.); herminia.mendivilal@gmail.com (H.M.-A.); 2Department of Chemical and Biological Sciences, Universidad de Sonora, Hermosillo 83000, Mexico; veronica.lopez@unison.mx (V.L.-T.); aegl.92@gmail.com (A.E.G.-L.)

**Keywords:** oncology, sarcopenic obesity prevention, body composition assessment, food-based individualized nutrition program

## Abstract

*Background and objectives:* Body composition assessment can provide information associated with breast cancer patients’ (BCP) prognosis, that can lead interventions to improve survival outcomes. The aim of this study was to evaluate the effect of an individualized nutrition intervention program on breast cancer patients using bioelectrical impedance vector analysis (BIVA). *Materials and Methods:* This is a pretest-posttest study in recently diagnosed nonmetastatic BCP undergoing antineoplastic treatment, free of co-morbidities and dietary supplementation. Body composition was assessed at baseline and 6 months after an individualized nutrition intervention program, by dual-energy X-ray absorptiometry and BIVA. According to BIVA, each participant was located in the bivariate tolerance ellipses for Mexican population (50%, 75%, and 95%). In clinical practice, the 50% and 75% ellipses are considered within normality ranges. *Results:* Nine nonmetastatic BCP completed the intervention and were included in the analysis. After the intervention, they decreased by 5.8 kg of body weight (IQR, 3–6; *p* < 0.05), 3.8 kg of fat mass (IQR, 0.1–4.2; *p* < 0.05), and 1.4 kg of fat-free mass (IQR, −0.1 to 4; *p* < 0.05) while appendicular skeletal muscle mass remained unchanged (−0.2 kg, IQR, −0.8 to 2.3; *p* = 0.4). Using BIVA at baseline, five participants were among the 50% and 75% ellipses, mainly located in the area corresponding to edema and low lean tissue, two in the cachexia quadrant and two in the athletic quadrant (≥95% ellipse). After 6 months of intervention, six out of nine participants were in the athletic quadrant and eight of nine BCP were above the 5° phase angle cut-off point. One patient initially presented cachexia (≥95% ellipse); at postintervention her vector changed to the 50% ellipse. *Conclusions:* An individualized nutrition intervention program designed for nonmetastatic BCP was effective to improve the nutritional status of BCP as assessed by BIVA, therefore BIVA can be a useful tool to monitor changes in nonmetastatic BCP body composition in research and clinical practice.

## 1. Introduction

During antineoplastic treatment, breast cancer patients (BCP) often experience an increase in body weight and fat mass (FM) and a decrease in fat-free mass (FFM), particularly skeletal muscle [1]. This condition is known as sarcopenic obesity (SO) and can affect BCP prognosis and quality of life [1,2]. Sarcopenia occurs in one out of three newly diagnosed patients and has been underrecognized in nonmetastatic BCP [3]. Measuring body constituents can provide valuable information associated with BCP prognosis that could lead interventions to improve survival outcomes [2,3]. Therefore, early nutritional and body composition assessment could be useful to consider in BCP management [2] for SO prevention.

Body composition assessment in BCP is essential to understand the effect of diet, disease, and other factors concerning the individuals’ health status [4,5]. Thus, most studies use reference methods to evaluate body composition [4], such as magnetic resonance imaging, computed tomography [2,3], or dual-energy X-ray absorptiometry (DXA) [1,4]. Yet, most of these methods are unaffordable in clinical practice [5], particularly in low- and middle-income countries with poor breast cancer survival [6]. Hence, precise, non-invasive and cost-effective methods are necessary to measure body composition determinants, especially in clinical practice [5].

Bioelectrical impedance analysis (BIA) based on a two-compartment model (FM and FFM) is one of the most commonly used methods to estimate body composition, due to its precision, low cost, and ease of application [5]. Since FFM estimations using BIA are linked to skeletal muscle and total body water, BIA can misclassify subjects with increased extracellular fluid volume as having an increase in their skeletal muscle compartment [7]. However, a novel approach can overcome BIA conventional limitations, called bioelectrical impedance vector analysis (BIVA), which is a qualitative method based on the analysis of a bivariate distribution of the impedance vectors in a healthy population. Individuals’ resistance (R) and reactance (Xc) values measured by BIA and standardized by height (H) are expressed in ohms/m and then each patient’s vector is located in an RXc graph according to his hydration status (edema or dehydration) and soft tissues content (lean and adipose) (Figure 1) [5,7,8].

The objective of this study was to evaluate the effect of an individualized nutrition intervention program using bioelectrical impedance vector analysis (BIVA), in nonmetastatic BCP under antineoplastic treatment.

## 2. Materials and Methods

### 2.1. Ethics, Study Design and Participants

The Ethics and Research Committees of the Research Center for Food and Development (CIAD) and the Oncology State Center (CEO), approved the study protocol and procedures on May 11th, 2015 (CE/005/2015) and May 2nd, 2016, respectively; the study was registered in ClinicalTrials.gov under the identifier NCT03625635. It is a non-randomized pretest-posttest study, conducted between March 2017–July 2018, in recently diagnosed nonmetastatic BCP undergoing antineoplastic treatment (surgery, chemotherapy, radiotherapy, and hormone therapy either combined or alone), at the Sonoran CEO. At baseline and before they received antineoplastic treatment, women with nonmetastatic breast cancer diagnosis, free of co-morbidities or use of dietary supplements were invited to participate. After informed consent was signed, clinical data were collected from their medical records.

### 2.2. Sample Size and Data Analysis

All recently diagnosed BCP who attended the CEO during March 2017–July 2018 were screened for eligibility criteria. Sample size was calculated based on a previous study conducted in the same institution, population, and 6-month duration as this study, where they evaluated the effect of antineoplastic treatment on Sonoran BCP body composition, without nutritional intervention [9]. Differences in BCP body weight were based the sample size calculation using the formula for a before-after study (paired *t*-test) with an independent continuous outcome:The standard normal deviate for α = Z_α_ = 1.960.(1)
The standard normal deviate for β = Z_β_ = 0.842.(2)
A = 1.000,(3)
B = (Z_α_ + Z_β_)^2^ = 7.849,(4)
C = (E/S(Δ))^2^ = 1.068,(5)
AB/C = 7.35,(6)
Group size N: 7 + 30% expected loss = 9 subjects required to complete the intervention.(7)

During the recruiting period, all 204 subjects who attended the CEO and were about to start antineoplastic treatment were contacted. Thirty participants met the inclusion criteria, but only 20 were enrolled in the study and 9 completed the 6-month nutrition intervention for quantitative and qualitative analysis, which meets the aforementioned sample size (Figure 2).

Data distribution is expressed in medians and interquartile ranges (IQR). Body composition constituents were analyzed using nonparametric Wilcoxon test for paired samples. A two-tailed *p*-value of 0.05 or less was considered significant. Data were processed using the statistical software NCSS^®^ 11.0 version.

BIVA qualitative results can be visualized as vectors in the RXc graph for Mexican population [5] and interpreted to diagnose hydration status and soft tissues content in steady state at baseline and vector migration postintervention (Figure 1) [5,8].

### 2.3. Body Composition Assessment

At baseline and 6 months after, weight was recorded in participants after ≥4 h of fasting during the morning on a digital scale (200 ± 0.05 kg AND^®^ model: HV-200KGL CC No. 00-088A) with light and metal-free clothes, without shoes or breast prosthesis. Standing height (H) was measured using a stadiometer (range of 62 to 209 cm ± 1 mm; SECA^®^ model: 242). Body composition was determined by DXA as the reference method (Hologic Corporation 4500 Waltham, MA, USA).

Resistance (R) and reactance (Xc) were measured by a BIA device at 50 kHz (Impedimed Limited DF50, Carlsbad, CA, USA). R and Xc were standardized by H expressed in ohms/m. To locate each participant in the bivariate tolerance ellipses for Mexican population, R/H and Xc/H were used (50%, 75%, and 95%). In clinical practice, the 50% and 75% ellipses are considered within normality ranges (Figure 1) [5].

### 2.4. Individualized Nutrition Intervention Program

Diet plans and recommendations were based on the individual’s socioeconomic and nutritional status, as well as treatment side effects (food affinity and tolerance) [4] using the dynamic macronutrient meal-equivalent menu method [10]. Resting energy expenditure was estimated using an algorithm for Mexican population [11], and when appropriate, a caloric restriction was considered [4]. WCRF/AICR guidelines [12] were followed adapting 1.5 g/kg/d of dietary protein [4]. Follow-up was every 2 weeks and a different diet menu was provided in each session by a specialized dietitian during the intervention until 6 months were completed.

## 3. Results

Nine women diagnosed with nonmetastatic breast cancer and undergoing antineoplastic treatment were included (Table 1).

At baseline, the mean age of participants was 44 ± 12 years. After the intervention, body size indicators (weight and BMI) decreased by 5.8 kg and 1.2 kg/m^2^ respectively (IQR, 3–6 and 0.8–2.4; *p* < 0.05). In terms of body composition by DXA, participants lost 3.8 kg of FM (IQR, 0.1–4.2; *p* < 0.05) and 1.4 kg of FFM (IQR, −0.1 to 4; *p* < 0.05) while skeletal muscle remained unchanged (−0.2 kg, IQR, −0.8 to 2.3; *p* = 0.4) (Table 2).

When BIVA was used at baseline, five participants were among the 50% and 75% ellipses, principally in the area corresponding to edema and low lean tissue, two in the cachexia quadrant and two in the athletic quadrant (≥95% ellipse). After 6 months of the nutrition intervention program, six out of nine participants were located in the athletic quadrant (two between 50% and 75% ellipses and the rest in the ≥95% ellipse). One of the two participants who initially presented cachexia (>95% ellipse), showed a change in her vector at the end of the intervention to the 50% ellipse, because of the favorable modifications in her body composition during this period (Figure 3).

At baseline, participants showed a 5.5° phase angle (PA), which changed to 7.6° after the nutrition intervention (+2.1°, IQR, −3.8° to 3.2°; *p* = 0.3). Nevertheless, it is noteworthy that eight out of nine participants after the intervention were above the 5° PA cut-off point estimated for better survival outcome (Table 1).

## 4. Discussion

To our knowledge, this is the first study conducted in nonmetastatic BCP during antineoplastic treatment with BIVA follow-up after a nutrition intervention program. There are considerable differences in healthy populations according to BIVA ellipses and PA reference values; therefore, using the Mexican population reference ellipses was the best option for our study group [5]. Further studies are needed in larger population samples considering lymphedema diagnosis [7,13], metastasis [14] and physical activity [1].

Skeletal muscle mass is one of the components of the fat-free mass compartment. Significant changes in FFM could reflect modifications in one of the components, such as hydration status [5]. This could contribute to the change in FFM rather than skeletal muscle mass or bone mineral content. However, changes in hydration status were detected by BIVA. Therefore, BIVA could potentially be used as a complimentary measuring tool to DXA.

Phase angle has been used as a survival predictor in different pathologies [15]. A PA greater than 5° was associated with longer survival rates in BCP [16] and other types of cancer [15,16,17,18], as presented in our volunteers. The only patient who reduced her PA at postintervention, was the one that remained in the cachexia quadrant (>95% ellipse). BIVA follow-up is at the moment the method that can detect these particularities.

### Limitations

Although the sample size in this study was small, it was based on the total population that met the inclusion criteria from the State of Sonora who attended the CEO, and proper non-parametric analyses were used to evaluate changes. Additionally, information in the patients’ medical record was limited and in most of them did not include lymphedema diagnosis. Since BIA may also be relevant in clinical practice by adding specificity and sensitivity in lymphedema detection, future studies should be conducted to assess and control this variable.

## 5. Conclusions

An individualized nutrition intervention program designed for nonmetastatic BCP was effective to improve the nutritional status of BCP as assessed by BIVA, therefore BIVA can be a useful tool to monitor changes in nonmetastatic BCP body composition in research and clinical practice.

## Figures and Tables

**Figure 1 medicina-55-00663-f001:**
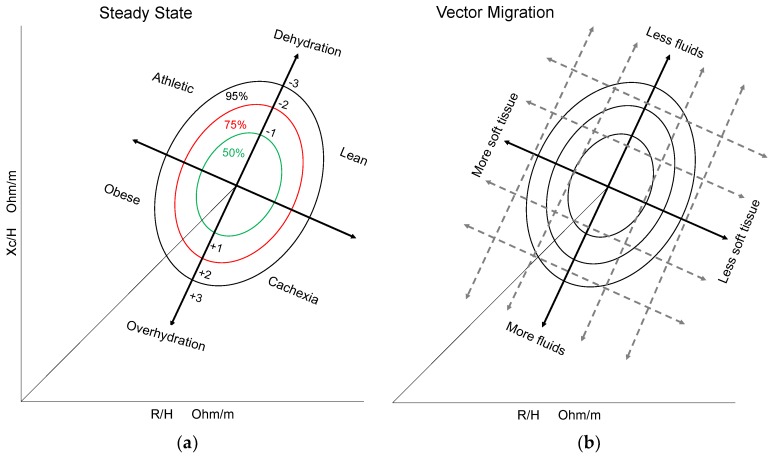
Bioelectrical impedance vector analysis (BIVA) steady state quadrants and vector migration according to hydration status and soft tissues content: (**a**) BIVA steady state quadrants; (**b**) BIVA vector migration. Optimal body composition is located at the center (50% and 75% ellipses). Vector migration to a certain direction represents a modification in the patient’s soft tissue/hydration status. Figure redrawn, adapted and modified [5,8].

**Figure 2 medicina-55-00663-f002:**
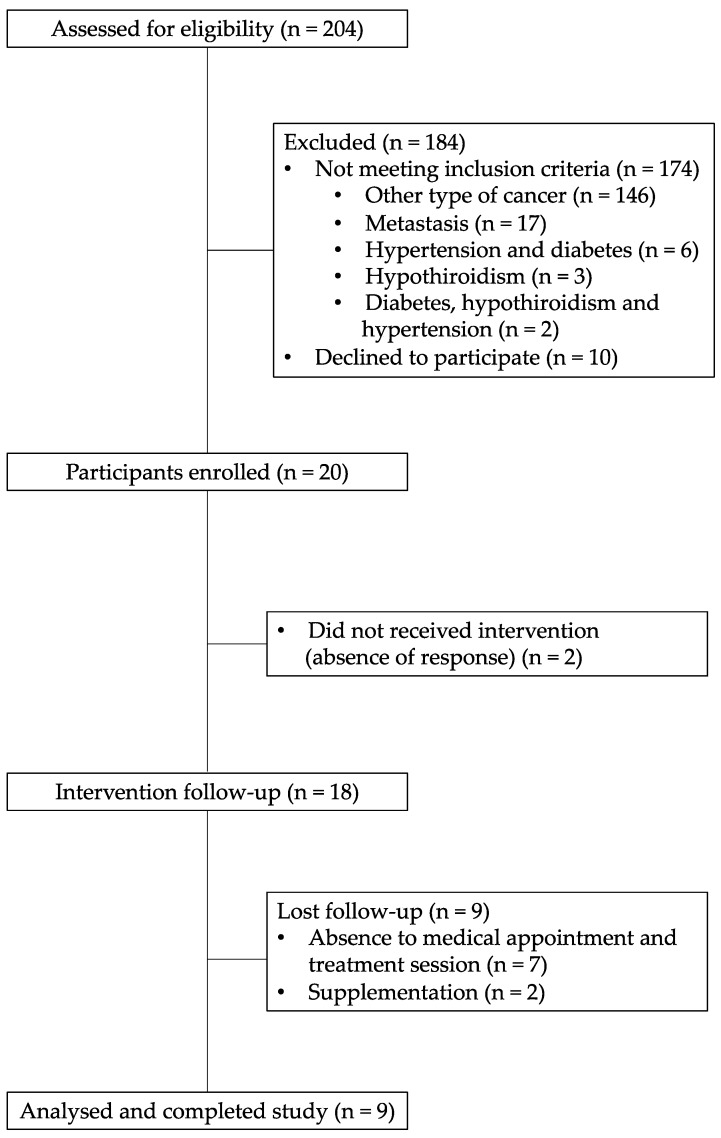
Consort diagram of study participants.

**Figure 3 medicina-55-00663-f003:**
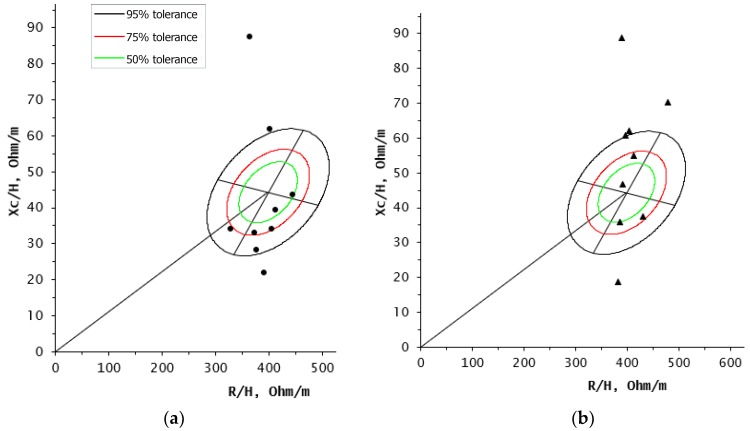
BIVA in breast cancer patients under antineoplastic treatment and after 6 months of a nutrition intervention (n = 9): (**a**) BIVA at baseline (●); (**b**) BIVA postintervention (▲).

**Table 1 medicina-55-00663-t001:** Patients treatment and clinicopathologic characteristics.

	*n* = 9
Surgery	
Quadrantectomy	4
Mastectomy	5
Breast cancer stage	
I	2
IIA	4
IIB	3
Histological subtype	
Invasive ductal carcinoma	7
Invasive lobular carcinoma	2
Molecular subtypes	
Luminal A	3
Luminal B	2
HER2	1
Triple Negative	3
Chemotherapy cycles	
0–6	3
7–8	6
Radiotherapy	
5000 cGy in 25 fractions	7
None	2

**Table 2 medicina-55-00663-t002:** Body composition in breast cancer patients at baseline and 6 months after the intervention.

	Baseline	Postintervention	Δ ^1^	*p* ^2^
	Median (IQR)			
Height, m	1.6 (0.1–0.2)	1.6 (0.1–0.2)	0	0.95
Body weight, kg	79.2 (10–27)	73.4 (13–22)	−5.8	<0.05
BMI ^3^, kg/m^2^	30.7 (7–11)	29.5 (7–9)	−1.2	<0.05
Fat mass, kg	33 (9–20)	29.2 (10–17)	−3.8	<0.05
Fat-free mass, kg	43 (5–8)	41.6 (7–9)	−1.4	<0.05
TASM ^4^, kg	15.5 (2–4)	15.3 (3–6)	−0.2	0.4
Resistance 50 kHz, ohm	639 (79–128)	639 (90–167)	0	0.01
Reactance 50 kHz, ohm	60 (35–103)	87 (45–117)	+27	0.2
Phase angle	5.5 (3–10)	7.6 (4–10)	+2.1	0.3

^1^ Δ = Postintervention – Baseline; ^2^ Wilcoxon test for paired samples (*n* = 9); ^3^ BMI: body mass index; ^4^ TASM: total appendicular skeletal muscle

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
