# Peer review of "Bioelectric Impedance Vector Analysis (BIVA) in Breast Cancer Patients: A Tool for Research and Clinical Practice"

_medicina, 2019, doi:10.3390/medicina55100663_

Round 1
Reviewer 1 Report
This study assesses the effect of an individualized nutrition intervention program to BCP under antineoplastic treatment by the more precise method, conventional method which you commented. Although the study has some merits like novelty, concise draft, it lacks actual informations in the methods section about a nutritional invention. And It needs to be announced a clinicopathologic charateristics about 9 patients like age, stage, and antineoplastic drug,etc.
Author Response
Dear editor:
The suggestions offered by the reviewers have been very helpful. We have included the reviewers’ comments and responded to them individually, indicating exactly how we addressed them and describing the changes we have made. The changes are marked with track changes in the revised manuscript, and all have been approved by all authors.
Additionally, we corrected references errors in the manuscript and included a sample size calculation in the analysis.
Reviewer 1.
This study assesses the effect of an individualized nutrition intervention program to BCP under antineoplastic treatment by the more precise method, conventional method which you commented. Although the study has some merits like novelty, concise draft, it lacks actual information in the methods section about a nutritional invention. And it needs to be announced a clinicopathologic characteristics about 9 patients like age, stage, and antineoplastic drug, etc.
Thank you for your input, we were aware of the limitation in the methods section that you describe. When we submitted this manuscript, the intervention’s methodology was being reviewed for publishing, fortunately the manuscript was approved, and we now address your concern inserting a new reference (1). Additionally, we included a table with the patients’ clinicopathologic characteristics in now Table 1 and age in line 184.
Limon-Miro AT, Lopez-Teros V, Astiazaran-Garcia H. Dynamic Macronutrient Meal-Equivalent Menu Method: Towards Individual Nutrition Intervention Programs. Methods Protoc [Internet]. 2019 Sep 5 [cited 2019 Sep 10];2(3):78. Available from: https://www.mdpi.com/2409-9279/2/3/78
Reviewer 2 Report
Questions I had during the review:
-What does antineoplastic treatment encompass (chemo, radiation, hormone therapy)? Would be helpful to clarify this.
-Was treatment regimen the same for all subjects and was this variation considered as a an influence to the results?
-Are the authors suggesting BIVA as a compliment to DXA, in replacement of DXA or as another method to evaluate/address sacropenic obesity?
-Any reflections on adequacy of nutrition intervention or was this out of scope of study?
-Fat-free mass had a statistically significant decrease but this was not discussed; any thoughts on this?
-Would this protocol be possible to apply to a large group of subjects?
Author Response
Dear editor:
The suggestions offered by the reviewers have been very helpful. We have included the reviewers’ comments and responded to them individually, indicating exactly how we addressed them and describing the changes we have made. The changes are marked with track changes in the revised manuscript, and all have been approved by all authors.
Additionally, we corrected references errors in the manuscript and included a sample size calculation in the analysis.
Reviewer 2.
Questions I had during the review:
-What does antineoplastic treatment encompass (chemo, radiation, hormone therapy)? Would be helpful to clarify this.
This is now clarified in the manuscript Lines 117-118 as: antineoplastic treatment (surgery, chemotherapy, radiotherapy, and hormone therapy, either combined or alone).
-Was treatment regimen the same for all subjects and was this variation considered as a an influence to the results?
Thank you for your input, we were aware of the limitation in the methods section that you describe. When we submitted this manuscript, the intervention’s methodology was being reviewed for publishing, and we now address the intervention’s characteristics including a new reference (1). Additionally, we included a table with the patients’ clinicopathologic characteristics.
Limon-Miro AT, Lopez-Teros V, Astiazaran-Garcia H. Dynamic Macronutrient Meal-Equivalent Menu Method: Towards Individual Nutrition Intervention Programs. Methods Protoc [Internet]. 2019 Sep 5 [cited 2019 Sep 10];2(3):78. Available from: https://www.mdpi.com/2409-9279/2/3/78
-Are the authors suggesting BIVA as a compliment to DXA, in replacement of DXA or as another method to evaluate/address sacropenic obesity?
DXA is one of the methods available in research to assess body composition and diagnose sarcopenic obesity, but in the clinical practice BIVA can be used as a complimentary measure. BIVA provides information on the hydration status of the patient and also a visual chart on the patient’s nutritional status compared to a reference population.
This is now included in the Discussion section lines 229-230.
-Any reflections on adequacy of nutrition intervention or was this out of scope of study?
Although this is crucial, it is fully addressed on a different manuscript (which is being finalized entitled: An individualized food-based nutrition intervention reduces visceral and total body fat while preserving skeletal muscle mass in breast cancer patients under antineoplastic treatment by Ana Teresa Limon-Miro, Veronica Lopez-Teros, Mauro E Valencia, Heliodoro Alemán-Mateo, Rosa O. Méndez-Estrada, Bertha I. Pacheco-Moreno and Humberto Astiazaran-Garcia) where the intervention and the impact on the nutritional status of breast cancer patients was the main scope. In the present manuscript we aimed to focus on the usefulness of BIVA as a tool to be applied in clinical practice for rapid assessment of the nutritional status in breast cancer patients. Included in the Discussion and Conclusion sections.
-Fat-free mass had a statistically significant decrease but this was not discussed; any thoughts on this?
Skeletal muscle mass is one of the components of the fat-free mass (FFM) compartment. Significant changes in FFM could reflect modifications only in the hydration status contributing to the change in FFM rather than skeletal muscle mass and bone mineral content (no significant change found), however, changes in hydration status were detected by BIVA. This is now briefly clarified in the discussion.
-Would this protocol be possible to apply to a large group of subjects?
This protocol is applicable to larger population groups.